# Polyaniline Modified CNTs and Graphene Nanocomposite for Removal of Lead and Zinc Metal Ions: Kinetics, Thermodynamics and Desorption Studies

**DOI:** 10.3390/molecules27175623

**Published:** 2022-08-31

**Authors:** Imran Ali, Tatiana S. Kuznetsova, Alexander E. Burakov, Irina V. Burakova, Tatiana V. Pasko, Tatiana P. Dyachkova, Elina S. Mkrtchyan, Alexander V. Babkin, Alexey G. Tkachev, Hassan M. Albishri, Wael Hamad Alshitari, Ahmed M. Hameed, Ahmed Alharbi

**Affiliations:** 1Department of Chemistry, Jamia Millia Islamia (Central University), Jamia Nagar, New Delhi 110025, India; 2Department of Chemistry, King Abdulaziz University, Jeddah 22254, Saudi Arabia; 3Technological Institute, Tambov State Technical University, 106 Sovetskaya St., 392000 Tambov, Russia; 4Federal State Research and Design Institute of Rare Metal Industry Giredmet, Electrodnaya St., 2 Building 1, 111524 Moscow, Russia; 5Department of Chemistry, College of Science, University of Jeddah, P.O. 80327, Jeddah 21589, Saudi Arabia; 6Department of Chemistry, Faculty of Applied Sciences, Umm Al-Qura University, Makkah 21955, Saudi Arabia

**Keywords:** polyaniline-CNTs-graphene nanocomposite, heavy metal ions removal, adsorption, kinetics, thermodynamics

## Abstract

A novel polyaniline-modified CNT and graphene-based nanocomposite (2.32–7.34 nm) was prepared and characterized by spectroscopic methods. The specific surface area was 176 m^2^/g with 0.232 cm^3^/g as the specific pore volume. The nanocomposite was used to remove zinc and lead metal ions from water; showing a high removal capacity of 346 and 581 mg/g at pH 6.5. The data followed pseudo-second-order, intraparticle diffusion and Elovich models. Besides this, the experimental values obeyed Langmuir and Temkin isotherms. The results confirmed that the removal of lead and zinc ions occurred in a mixed mode, that is, diffusion absorption and ion exchange between the heterogeneous surface of the sorbent containing active adsorption centers and the solution containing metal ions. The enthalpy values were 149.9 and 158.6 J.mol^−1^K^−1^ for zinc and lead metal ions. The negative values of free energies were in the range of −4.97 to −26.3 kJ/mol. These values indicated an endothermic spontaneous removal of metal ions from water. The reported method is useful to remove the zinc and lead metal ions in any water body due to the high removal capacity of nanocomposite at natural pH of 6.5. Moreover, a low dose of 0.005 g per 30 mL made this method economical. Furthermore, a low contact time of 15 min made this method applicable to the removal of the reported metal ions from water in a short time. Briefly, the reported method is highly economical, nature-friendly and fast and can be used to remove the reported metal ions from any water resource.

## 1. Introduction

Water is the basis of life on our planet, ensuring the existence of humanity. Various natural and anthropogenic factors can lead to the contamination of water resources with toxic pollutants, among which heavy metal compounds are the most hazardous and carcinogenic [1,2,3,4,5]. The growing technological impact of modern industrial enterprises and factories on the quality and purity of water contributes to the active growth of research on the effective removal of these contaminants from aqueous media using adsorption methods [6]. In recent decades, nano materials are gaining importance in many fields [7,8,9]. In particular, carbon nanotubes (CNTs), due to their highly developed surface, low mass density, effective porosity and ease of modification, are widely used to remove various heavy metals [8]. At the same time, an increase in the accessibility of the CNT surface is facilitated by their dispersed stability through oxidation. For example, commercial single-walled and multi-walled carbon nanotubes (CNTs) were used to purify water from zinc ions Zn^2+^ [9]. To impart hydrophilicity, the CNTs were heat-treated at 350 °C for 30 min, cleaned with 70% sodium hypochlorite solution and subjected to ultrasonic cleaning for 20 min in a water bath (at 85 °C). The adsorption capacity for Zn (II) was 43.66 mg/g.

Carbon nanotubes oxidized with H_2_O_2_, KMnO_4_ and HNO_3_ were used to adsorb Cd (II) ions [10]. Due to the functional groups introduced by oxidation, the adsorption capacity of Cd (II) reached 2.6, 5.1 and 11.0 mg g^−1^ for CNTs oxidized with H_2_O_2_, HNO_3_ and KMnO_4_, respectively, at an equilibrium concentration of Cd (II) of 4 mg L^−1^. The specific surface area and specific pore volume of CNTs increased after the oxidation of H_2_O_2_, HNO_3_ and KMnO_4_. Also, dispersions of oxidized multi-walled CNTs (o-CNTs) were used to remove heavy metals from acidic solutions [11]. Oxidized multiwalled CNTs (in 65% nitric acid) were dispersed in deionized water using ultrasound in the presence of various surfactants. Numerous studies have noted that surface modification of CNTs leads to an increase in the adsorption capacity of negatively charged surfaces of modified CNTs [12]. An even higher sorption capacity was demonstrated by a 2D carbon material. Melezhik et al. [13] coated graphene oxide (GO) with various oxygen-containing groups and reported a high surface area. Zhao et al. [14] described 106.3 and 68.2 mg/g sorption capacity of Cd (II) and Co (II) on graphene oxide at pH 6.0 ± 0.1 and T = 303 K. Wang et al. [15] used ethylenediamine as a reducing agent and the obtained GO showed an adsorption capacity of 413.22, 162.33, 55.34 and 42.46 mg/g for Pb (II), Cd (II), Cu (II) and Mn (II), respectively. GO reduced by thiourea dioxide was studied for the adsorption of Th (IV) from aqueous solutions by Pan et al. [16]. The maximum theoretical adsorption capacity of Th (IV) was 0.21 mmol at 298.15 K. It was also shown that the adsorption of Th (IV) on r-GO was described by pseudo-second-order and Langmuir isotherm models. In addition, thermodynamic parameters were determined (Δ*H*_0_ = 4.32 kJ/mol, Δ*S*_0_ = 69.47 J/mol K and Δ*G*_0_ = −16.39 kJ/mol at 298.15 K), which showed that the adsorption process was spontaneous and endothermic in nature.

An important advantage of r-GO is the possibility of chemical modification of its surface without damage to the bulk and structural properties of the material, which made it possible to noticeably increase the adsorption capacity. As a result, r-GO is a more promising 2D material than the original GO due to the possibility of compacting and forming various commercial adsorbents for use in aqueous media. Polyaniline (PANI) is a very promising modifying agent, which contains alternating iminoquinoid and phenylenediamine groups. This is capable of adsorbing cationic and anionic contaminants through various electrostatic interactions [17,18]. The combination of PANI with carbon nanomaterials made it possible to adsorb heavy metal ions. For example, GO obtained by polymerization at temperatures below −20 °C with PANI nanodots showed a high Cr (VI) adsorption capacity of 1149.4 mg g^−1^ in combination with the reduction to Cr (III). Another three-dimensional PANI/GO composite showed 80.7 mg g^−1^ at pH 5 adsorption capacity for Hg (II). In general, it was observed that PANI/GO composites absorb heavy metals efficiently, but most publications focus on the most common Zn (II), Cd (II), Pb (II) and Cu (II) ions. Hajjaoui et al. [19] reported good removal capacities of heavy metal ions by modifying multilayer CNTs with polyaniline. However, the use of PANI is limited by its extremely low solubility in water and most known solvents, as well as poor mechanical properties—the polymer powder does not adhere to other materials. One of the ways to eliminate this disadvantage is to obtain composite systems that combine the effective functional properties of PANI and the dispersion stability of the carrier adsorbent [20]. There is a method for obtaining an aqueous colloidal solution of PANI, which can be used, in particular, to obtain composite graphene materials [21]. To obtain a colloidal solution, the polyaniline base is sonicated in an aqueous solution of a resole-type phenol-formaldehyde resin. It is reasonable to assume that even more effective adsorbents can be nanocomposites in which a mixture of carbon nanomaterials is modified with polyaniline. The possibility of forming such a composite (GNS/CNT/PANI) is mentioned by Rahman et al. [22]. Polyaniline (PANI) is a famous conductive polymer, and it has received tremendous consideration from researchers in the field of nanotechnology for the improvement of material properties including sorbents. PANI is doped easily because of its easy synthesis and remarkable environmental stability. PANI is one of the most effective organic polymeric materials used to remove various cationic and anionic pollutants from aqueous media, due to the significant content of phenylenediamine and quinonediamine functional groups. Due to its high nitrogen functionality, this polymer is often used in the extraction of heavy metal ions from aqueous media [23]. Keeping all these facts into consideration, this article implements the synthesis of a hybrid nanostructured adsorbent based on r-GO modified with PANI. In this method, o-CNTs played a role of a structure-forming agent and as an additional component that improved the efficiency of zinc and lead metal ions removal. The developed functional nanomaterial is a multicomponent cryogel obtained using the lyophilization method.

## 2. Materials and Methods

### 2.1. Materials

To prepare a nanocomposite (cryogel r-GO/o-CNTs/PANI), the substances and materials used were oxidized CNTs (o-CNTs) obtained by oxidation of Taunit-M CNTs (NanoTechCenter LLC, Tambov, Russia). Besides this, a solution of sodium hypochlorite in the form of an aqueous paste with a mass content of 11.48% dry matter; r-GO (NanoTechCenter LLC, Tambov, Russia; prepared by reduction of graphene oxide with ascorbic acid); acetic acid (1.5 M CH_3_COOH) (Nevinnomyssky AZOT JSC, Nevinnomyssk, Russia); resole water-soluble PF Fenotam-GR-326 (Krata PJSC, Tambov, Russia; with a dry matter content of 50%; aniline hydrochloric acid; ammonium persulfate; aqueous ammonia containing 25% ammonia and hydrochloric acid (JSC REACHIM LLC, Moscow, Russia) were used. To determine the adsorption capacity of the cryogel for heavy metals, model solutions of nitrates were used, that is, zinc nitrate Zn(NO_3_)_2_·6H_2_O and lead (II) nitrate Pb(NO_3_)_2_ (Scientific and Production Kama Chemical Company LLC, Perm, Russia).

### 2.2. Preparation Methods

#### 2.2.1. Polyaniline Preparation Method

PANI was produced under laboratory conditions by the standard method, including the oxidative polymerization of aniline in an acidic medium [24]. In 0.115 M HCl 97.2 g (0.75 mol) of aniline hydrochloride was added and stirred until complete dissolution. Separately, an aqueous solution was prepared of 213.93 g (0.9375 mol) of ammonium persulfate in 375 mL water, which was then added to the hydrochloric acid solution of aniline hydrochloride with continuous stirring in an argon atmosphere, after which the stirring process was continued for another 2 h. Then the mixture was kept for 11 h to complete the polymerization process. Then the reaction mixture was transferred to a filter and washed first with a 0.1 M hydrochloric acid solution until a colorless filtrate was obtained following washing with distilled water until pH ≈ 6. Further, for deprotonation, the product was washed on the filter with a 1 M ammonia solution and then with distilled water until the filtrate had a neutral pH. Excess moisture was filtered out using a vacuum pump, and the finished product was packed in a sealed container. To determine the mass content of dry matter in the resulting paste, a portion of the sample was dried in an oven at 110 °C to constant weight. The mass content of dry matter was 11%. The final material was used in the form of an aqueous paste since complete drying leads to a strong and irreversible agglomeration of the particles.

#### 2.2.2. Nanocomposite Preparation Method

Synthesis of the base hydrogel was carried out at room temperature without preliminary preparation of materials. Aqueous pastes of the initial components (r-GO/o-CNTs/PANI/FFS) were taken in an amount corresponding to 1 g of dry matter, mixed in a ratio of 1:1:1:1 in distilled water, bringing the total weight to 100 g. Then, the mixture was subjected to ultrasonication for an hour (frequency 22 ± 0.4 kHz) to destroy particle aggregates and uniformly distribute the mixture components. Then, 25 mL of 9% acetic acid was added to the resulting colloid to coagulate the solution. The mixture was kept for 1 h to complete coagulation and then filtered on a polypropylene microfilter and washed with distilled water to remove reaction side-products. Excess moisture was removed with a vacuum filter for 10 min. To prepare a dry r-GO/o-CNTs/PANI nanocomposite, the solvent was removed from the hydrogel pores by freeze-drying at a temperature of −40…−50 °C and a pressure of about 5–8 Pa using a Scientz-10n unit (Science, Ningbo, China).

### 2.3. Characterization Methods

Micrographs of the materials were obtained using a scanning electron microscope (SEM) MERLIN (Carl Zeiss, Jena, Germany) and a transmission electron microscope (TEM) JEM-2010 instrument (JEOL, Tokyo, Japan). The structural characteristics of the initial components and the nanocomposite were analyzed by X-ray phase analysis using a Thermo Scientific ARL Equinox 1000 X-ray diffractometer (TechTrend Science Co., Ltd., Taoyuan, Taiwan). The diffraction patterns were taken at a wavelength of λ = 0.1540562 nm (copper anode). Data registration was carried out in the range of 2θ angles from 10° to 110°. Using the diffraction patterns, the structure of crystalline substances and phase composition were studied. The sizes of crystallites (regions of coherent scattering) were determined by the Debye–Scherrer formula (1) [25]
(1)$$D=\frac{n\cdot \lambda }{\beta \cdot \cos\theta }$$
where *D* is crystallite size, nm; *n* is the coefficient depending on the particle shape and close to 1; λ is the wavelength of X-ray radiation, nm; β is the half-width of the peak in the diffraction pattern, rad; θ is peak maximum angle, deg.

To study the dependence of the change in mass with temperature, a combined thermal analysis was carried out involving the simultaneous use of thermogravimetric analysis (TGA) and differential scanning calorimetry (DSC) using a NETZSCH STA 449 F3 Jupiter instrument. The study was carried out in air at a heating rate of 10 K/min. The specific surface area and pore space parameters were measured by nitrogen adsorption at 77 K using an Autosorb iQ automatic analyzer (Quantachrome Instruments). The multipoint Brunauer–Emmett–Teller (BET) method was used to determine the specific surface area, and a mathematical model based on the density functional theory (DFT) was used to determine the volume and pore size. The degassing procedure was carried out at 200 °C and a pressure of 1.3 × 10^−3^ Pa for 8–10 h. For calculating the specific surface area, P/P0 values up to 0.3 were used. Samples were analyzed using a DXR Raman Microscope with an excitation laser wavelength of λ = 633 nm and a range of 100–3500 cm^–1^. To obtain the FT-IR spectra of the initial components and the nanocomposite, a Jasco FT/IR 6700 IR Fourier spectrometer was used. FTIR measurements were performed using KBr pellets. Number of scans—16. Resolution—4 cm^−1^. Samples were prepared for analysis. The aqueous pastes and suspensions were dried in an oven at 50 °C to constant weight and pressed into tablets with potassium bromide.

### 2.4. Adsorption Study

The equilibrium concentration of heavy metal ions in selected aliquots of aqueous solutions (deionized water) was measured using energy dispersive X-ray fluorescence spectrometry (ARLQuant ThermoScientific spectrometer, ThermoScientific, Waltham, MA, USA).

Static sorption capacity of sorbents $q_{e}$, mg g^−1^ was calculated by the formula:(2)$$q_{e}=\frac{\left(C_{0}-C_{e}\right)\cdot V}{m}$$
where *C*_0_ and *C_e_* are initial and final concentrations of substances in solution, mg L^−1^; *V* is the solution volume, L; *m* is the weight of sorbent sample, g.

#### 2.4.1. pH Effect

The study of the effect of the solution pH on the sorption capacity of the r-GO/o-CNTs/PANI nanocomposite was carried out using an H1 2210 pH Meter. A buffer system with a given pH value was gotten by adding the required acids (hydrochloric, acetic and amino acetic) and alkali (NaOH) to deionized water in given proportions. The experiments were carried out at room temperature. Stock solutions were obtained using zinc nitrate [Zn(NO_3_)_2_·6H_2_O] and lead (II) nitrate [Pb(NO_3_)_2_] of analytical grade by dissolving weighed portions in 1 L of deionized water, which were then diluted to the required concentrations. The sorbent weighing 0.005 g was placed in conical tubes with a capacity of 50 mL and poured into 30 mL of a model solution of heavy metals with *C*_0_ = 100 mg L^−1^. The tubes were shaken for an hour on a rotator, then the solution was filtered to distinguish the solid phase. All experimental situations for the process were stable for each sample.

#### 2.4.2. Dosage Effect

For precise valuation of the sorption activity of the material, it is essential to use the weight of the adsorbent sample at which the sorption ability for each extractable component is maximum under the designated experimental situations of the process. To regulate the effect of the weight of the adsorbent on its sorption capacity, samples were used, which are acetic acetate buffer systems with a volume of 30 mL, containing solutions of Zn(NO_3_)_2_·6H_2_O and Pb(NO_3_)_2_ with *C*_0_ = 150 mg L^−1^. For experimental studies, the sorbent weights used were 0.001, 0.005, 0.01, 0.02, 0.03 and 0.05 g. The tubes were stunned for 30 min on a rotator, then the solution was cleaned to separate the solid phase, and the equilibrium concentration of ions in the solution was measured.

#### 2.4.3. Kinetic Study

To determine the optimal sorption time of Pb (II) and Zn (II) ions on the synthesized nanocomposite, kinetic experiments were carried out in a limited volume (batch kinetics experiments). The sorbent weighing 0.005 g was placed in 30 mL of heavy metal solution with *C*_0_ = 100 mg L^−1^. Given the significant sorption activity in the initial period of the process, it is necessary to determine a larger number of samples in the preliminary period of time. The experiments were carried out at 5, 10, 15, 30, 45 and 60 min time intervals. Then the solution was filtered to discrete the solid phase, and the equilibrium amount of ions in the solution was decided.

#### 2.4.4. Isotherm Study

The study of the state of adsorption equilibrium was carried out using samples of model solutions of metal ions with varying concentrations at a sorbent dose of 0.005 g. A solution of the required concentration was made by sequential dilution of acetic acetate buffer solutions with an amount of 300 mg L^−1^ (solutions with *C*_0_ = 20, 50, 100, 200 and 300 mg L^−1^ were obtained). The tubes were shaken in a rotator for 30 min and filtered to discrete the solid phase. Then the equilibrium amount of ions in the solution was measured.

#### 2.4.5. Thermodynamic Study

To confirm the correctness of the expectations around the nature of the studied sorption procedures on the made nanocomposite, it is required to examine the effect of temperature on the efficacy of ion removal from aqueous solutions. For this, 30 mL solutions of various concentrations *C*_0_ = 20, 50, 100, 200, 300, 400 mg L^−1^ were used. Sorbent in the amount of 0.005 g was added to the solution and then shaken for 30 min. In this case, the isotherms were obtained at different temperatures: 25, 35 and 45 °C. The solution was filtered to remove the solid, and the equilibrium amount of ions in the solution was measured.

#### 2.4.6. Desorption Study

To study the effect of desorption, 30 mL of a solution with a given concentration and 0.005 g of a sorbent were taken. The initial solution was prepared by successive dilution of acetic-acetate buffer solutions of lead and zinc ions with *C*_0_ = 400 mg L^−1^ (solutions were obtained *C*_0_ = 20, 50, 100, 200, 300, 400 mg L^−1^). The tubes were shaken on a rotator for 30 min, and the filtrate was taken to measure the equilibrium concentration. Then, after separating the solid phase, it was dried at a temperature of 55 °C for ~15 min to remove surface moisture. After that, the dried nanocomposite was poured with a 30 mL buffer solution with pH = 6 and shaken for an hour on a programmable rotator. The suspension was then filtered and the equilibrium concentration was measured after the desorption stage.

### 2.5. Process Modeling

#### 2.5.1. Adsorption Kinetics

To describe kinetics, models are traditionally used that simulate the processes of mass transfer using formal equations of chemical kinetics. This is due to the complexity of the quantitative description of diffusion processes using simple models. This approach most often involves the use of pseudo-first- and pseudo-second-order models [26,27].

The pseudo-first-order equation of the Lagergren model has the following form.
(3)$$\frac{dq_{t}}{dt}=K_{1}\left(q_{e}-q_{t}\right)$$
where $q_{e}$ is adsorption at equilibrium, mg g^−1^; $q_{t}$ is adsorption at time t, mg g^−1^; $K_{1}$ is pseudo-first-order adsorption rate constant, min^−1^.

After integration and application of boundary conditions (from t=0  to t=t and $q_{t}=0$ to and $q_{t}=q_{t}$) the equation takes the following form.
(4)$$log\left(q_{e}-q_{t}\right)=log\;q_{e}-\frac{K_{1}}{2.303}t$$

It should be noted that the equation of the pseudo-first-order kinetics model coincides mathematically with the equation characteristic of diffusion processes. However, the value of the sorption rate constant was used here as a fixed parameter of the model. In the case of diffusion, the rate of the process depends on the size of the sorbent particles and the film thickness. If the limiting stage is chemical kinetics, then the sorption rate does not depend on the above factors but depends only on the concentration and temperature. Thus, when the sorption process kinetics is described by a pseudo-first-order kinetics model, the sorption of a substance is preceded by its diffusion.

The classical form of the equation of the pseudo-second-order kinetics model [28,29] has the following form.
(5)$$\frac{dq_{t}}{dt}=K_{2}{\left(q_{e}-q_{t}\right)}^{2}$$
where $K_{2}$ is pseudo-second-order adsorption rate constant (g mg^−1^ min^−1^).

After integration and application of boundary conditions (from t=0  to t=t and $q_{t}=0$ to and $q_{t}=q_{t}$) the equation takes the following form.
(6)$$\frac{t}{q_{e}-q_{t}}=\frac{1}{q_{e}}+K_{2\;}t$$

The linearized form of the equation that is most commonly reported in the literature has the following form.
(7)$$\frac{t}{q_{t}}=\frac{1}{K_{2\;}q_{e}^{2}}+\frac{1}{q_{e}}t$$

The applicability of this model indicated the occurrence of a chemical reaction that limits the sorption kinetics. Azizian [30] and Liu and Shen [31] showed that Langmuir kinetics could be reduced to either pseudo-first-order velocity equations or second-order velocity equations. However, there are works in which the pseudo-second-order model is considered a rather flexible mathematical formula capable of modeling the sorption kinetics of internal mass transfer (diffusion) for systems with flat and spherical particles [32].

The Elovich model is mainly used to describe chemical adsorption on highly heterogeneous adsorbents, but it does not suggest any specific mechanism for the adsorbate–adsorbent interaction [33].

The equation of the Elovich model has the form:(8)$$q_{t}=\frac{1}{\beta }ln\left(\alpha \beta \right)+\frac{1}{\beta }lnt$$
where *α* is the preliminary adsorption rate constant (min^−1^ mg g^−1^); *β* is the surface coverage and activation energy of chemisorption (g mg^−1^).

Weber and Morris described the intraparticle [34] uptake of the adsorption process to be proportional to the half-power of time:(9)$$q_{t}=k_{id}t^{0.5}+C$$
where *k_id_* is the internal diffusion coefficient (mg g^−1^min^−0.5^); *C* is the boundary layer thickness (mg g^−1^).

#### 2.5.2. Adsorption Isotherms

The Langmuir equation deals with monolayer adsorption [35]. Adsorption occurs on a homogeneous surface covered with adsorption sites. The free centers of the sorbent and the adsorbate form an interaction in an ideal solution.

The linear form of the Langmuir isotherm has the following form:(10)$$\frac{C_{e}}{q_{e}}=\frac{1}{K_{L}\cdot q_{max}}+\frac{C_{e}}{q_{max}}$$
where $q_{max}$ is maximum sorption capacity, mg g^−1^; $K_{L}$ is adsorbent–adsorbate affinity distribution (adsorption energy), L mg^−1^.

The Freundlich isotherm is one of the earliest empirical equations used to describe non-ideal sorption equilibrium data and describes polymolecular adsorption. According to Freundlich’s theory, the absorption process is most often initial surface adsorption followed by condensation resulting from strong interactions [35].

The linear form of the Freundlich isotherm is given below.
(11)$$lnq_{e}=lnK_{F}+\frac{lnC_{e}}{b_{F}}$$
where $K_{F}$ and $b_{F}$ are Freundlich constants, $b_{F}\;$ giving an indication of how favorable the adsorption process and $K_{F}$ (mg g^−1^ L mg^−1^) is the adsorption capacity of the adsorbent.

The Temkin isotherm contains a parameter that takes into account interactions between adsorption sites and the adsorbate. The model assumes that the heat of adsorption of all molecules in the layer decreases linearly as the layer is filled. This process proceeds to a greater extent according to a linear law than according to a logarithmic one [36].

The linear form of the Temkin isotherm is given below.
(12)$$q_{e}=B_{T}lnK_{T}+B_{T}lnC_{e}$$
(13)$$B_{T}=\frac{RT}{b}$$
where $K_{T\;}$ is equilibrium binding constant corresponding to the maximum binding energy (L.mg^−1^) $B_{T}$ is the Temkin isotherm constant. T is an absolute temperature (K). R is a universal gas constant (8.314 J (mol K)^−1^). b is a constant related to the heat of sorption (J.moL^−1^).

#### 2.5.3. Adsorption Thermodynamics

To determine the spontaneity of the adsorption process and describe the absorption mechanism, enthalpy change (ΔH°), entropy change (ΔS°) and free energy change (ΔG°) are measured [37].

The Gibbs equation is used to study the effect of temperature on equilibrium adsorption. The equation can be represented as follows:(14)$$\Delta G{}^{\circ}=-RTlnK_{c}$$

$K_{c}$ is a thermodynamic equilibrium constant (L mol^−1^), which can be found through the following equation:(15)$$K_{c}=\frac{C_{Ae}}{C_{Se}}$$
where $C_{Ae}\;$ is the equilibrium concentration of the adsorbate adsorbed on the sorbent (mg L^−1^) and $C_{Se}$ is an equilibrium concentration of adsorbate in solution (mg L^−1^).

The value of the enthalpy of the sorption process is found by graphical interpolation of experimental data using the Van’t Hoff isotherm equation.
(16)$$lnK=-\frac{\Delta H}{RT}+C$$

#### 2.5.4. Desorption

To calculate the desorption, we used the following equation.
(17)$$R_{des}=\frac{C_{e}^{des}}{\left(C_{0\;}-C_{e\;}\right)}$$
where $C_{e}^{des},C_{0\;,\;}C_{e\;}\;$ are equilibrium concentrations in solution during desorption, initial concentration of the solution and equilibrium concentration in solution during adsorption, respectively [38].

## 3. Results and Discussion

### 3.1. Characterization

#### 3.1.1. Scanning and Transmission Electron Microscopy

Figure 1 shows SEM and TEM images of the surface structure of a freeze-dried cryogel. CNTs are randomly distributed over the surface of graphene planes, which are uniformly coated with a layer of polyaniline.

The presented SEM and TEM images enable us to conclude that all the obtained samples have an ordered porous structure. In this case, CNTs act as a structure-forming agent and a supporting frame that prevents the aggregation of graphene sheets while maintaining the target functional properties of materials. PANI uniformly coats reduced graphene oxide and CNT sheets.

#### 3.1.2. XRD Analysis

Figure 2 shows X-ray diffraction patterns of the initial components and the nanocomposite. It is known that material with well-crystallized and uniform lattice parameters gives narrow diffraction peaks, while a poorly crystallized inhomogeneous material produces wide ones. For crystallites (coherent scattering regions) of small size, the diffraction lines are broadened. The half-width of the diffraction peak is larger than the smaller size of the coherent scattering regions (CSRs).

Table 1 shows the values of interplanar distances (d) for initial materials.

PANI is a semi-crystalline polymer and can crystallize into phases with triclinic and monoclinic lattices, characterized by peaks at 2θ = 15°, 20° and 25° in a diffraction pattern. An increase in the crystallinity leads to an increase in the ordering of the structure and an increase in charge mobility between polymer chains. The diffraction pattern of polyaniline shows a peak at 20°, which corresponds to a periodicity along the polymer chain. This is explained by the fact that; in the deprotonated form of PANI (base); the periodicity across the chains was practically lost and only a peak with periodicity along the chains remained [39,40]. For r-GO and CNTs, three peaks at 25°, 42° and 77° 2θ values are observed in the region of intense diffraction. These peaks at 2θ = 25° correspond to (002) planes (basic planes) of graphite. The broader peak centered on 2θ in the 44° region originated from (101) planes. The peak at 2θ in the 78° region was of halo and originated from (110) planes [41]. A nanocomposite is a material having a layered structure with an interlayer distance close to the interlayer distance of graphene layers in graphite. After freeze-drying (Figure 2), the ordered structure of the carbon framework of the materials is preserved. The material contains reflections characteristic of the original substances (Table 2).

The sizes of the crystallites (regions of coherent scattering), calculated by the Debye–Scherrer formula, for angles 2θ in the region of 19.11° amounted to 2.32 nm; for 25.22° —3.55 nm; for angles 2θ in the region of 42.59°—7.34 nm.

#### 3.1.3. Thermal Analysis and Differential Scanning Calorimetry

Figure 3 shows the TG/DSC curves obtained for the nanocomposite under study.

From the presented data, it can be seen that in the temperature range of 50–300 °C, the weight loss was ~ 9.6%, which can be associated with the elimination of water or other low molecular weight components. In the temperature range of 300–550 °C, the main weight loss occurred at about 89%, probably due to thermal-oxidative degradation [42].

#### 3.1.4. Analysis of Specific Surface Area, Volume and Pore Size

Figure 4 shows the results of diagnostics of the porous structure for samples obtained from the base hydrogel by freeze-drying (lyophilization).

The specific surface of the cryogel calculated by the BET method was 176 m^2^ g^−1^. The specific pore volume was 0.232 cm^3^ g^−1^. In addition, the material mainly contains pores with a size of 1.5–4 nm.

#### 3.1.5. Raman Spectroscopy

Figure 5 shows the Raman spectra of the initial components and the synthesized nanocomposite. The spectrum of polyaniline is typical for the typical emeraldine basic form of PANI. The intense band at 1590 cm^−1^ and the relatively narrow band at 1166 cm^−1^ are due to the stretching and bending vibrations of the C-C bonds in the quinoid ring, respectively. The dominant peak in the spectrum at 1470 cm^−1^ and a fairly well-defined peak at 1218 cm^−1^ are associated with the presence of C-N bonds. A number of bands at 840, 780 and 747 cm^−1^ are due to bond vibrations in substituted aromatic rings, including in phenazine-like fragments. Two bands at 525 and 416 cm^−1^ refer to out-of-plane deformations of the benzene ring [43]. The spectrum of CNTs and r-GO contains two characteristic modes: G (1500–1600 cm^–1^), caused by vibrations of carbon atoms in the plane of the graphene layer in the state of sp^2^-hybridization and D (1250–1450 cm^–1^), associated with the presence carbon atoms in the state of sp^3^-hybridization, which occurred when topological defects appeared in graphene layers or associated with the presence of amorphous carbon particles [44]. The second-order scattering spectrum (2400–3200 cm^−1^) was represented by wide low-intensity bands at ~2700 (G′) and 2950 (D + D′) cm^−1^, which are characteristic of oxygen-containing functional groups [41].

The peak intensity ratio I_D_/I_G_ is used to assess the defectiveness of the CNT surface. During oxidation, new defective areas may appear on the surface of graphene layers due to the formation of functional groups. At the same time, the I_D_/I_G_ indicator should increase. Thus, the I_D_/I_G_ ratio for CNTs, r-GO and nanocomposite were 1.14, 1.43 and 1.22. The cryogel spectrum (Figure 5) contains all the peaks characteristic of the original components. This fact confirmed the presence of both carbon nanostructures and PANI in the nanocomposite.

#### 3.1.6. Infrared Spectroscopy

The IR spectrum of r-GO/o-CNTs/PANI nanocomposite cryogel (Figure 6) contains peaks due to the presence of various functional groups and bonds of carbon atoms with them. Peaks at 2982, 2943, 2865 cm^−1^ are caused by vibrations of C-H bonds in alkyl groups [45]. The same peaks are usually found in the spectra of CNTs and GO [46,47].

The peak at 1627 cm^–1^ is caused by vibrations of C=C bonds in aromatic rings [48,49,50], which are structural units of the graphene planes of the initial carbon nanomaterials, as well as fragments of PANI and PFC molecules. Various oxygen-containing groups are present on the surface of the resulting nanocomposite. Thus, the peak at 3400 cm^–1^ is due to vibrations of O–H bonds in hydroxyl groups [48,50]. The peaks at 1733 and 1167 cm^–1^ are due to vibrations of C=O and C–O bonds, respectively. The same groups are found in GO. The presence of PANI fragments in the nanocomposite is confirmed by peaks at 1468 and 1313 cm^–1^ caused by the vibrations of the C=N bonds in secondary aromatic amine, C–N in the secondary aromatic amine [48,51], as well as a peak at 822 cm^–1^ due to N-H bond The prominent 1137 cm^–1^ band is assigned to a vibration mode of –NH^+^– structure [52]. It can also be assumed that the broad-band in the region 3100–3500 cm^−1^ contains an additional peak at 3300 cm^−1^, which is characteristic of N-H bonds. Electron-donating oxygen and nitrogen-containing groups, which are identified according to IR spectroscopy data in the composition of the nanocomposite, can be active centers for the adsorption of heavy metal ions due to the possibility of forming coordination bonds (Figure 6), similarly to how it was shown in [53].

### 3.2. Determination of the Sorption Characteristics of the Samples

#### 3.2.1. pH Test

The optimization of the pH of water is important in the adsorption process. Therefore, the experiments were carried out from 2.0 to 9.0 pH values. The pH value of the solution has a significant effect on the efficiency of the sorption interaction of the contaminant with the sorbent.

The results showed that the maximum adsorption capability of the nanocomposite was attained at pH = 6.5. As can be seen from the presented graph (Figure 7), the dependences of adsorption on the pH of the solution have an extreme character. Relatively low absorption rates in an acidic medium at pH ~ 2–4 are related to a high content of competing hydronium ions. In the range of pH values from 3 to 6, there is an increase in the capacity of the sorbent with a maximum at pH = 6. In this area, the amount of H^+^ hydroxonium ions was maximum reduced, which intensified sorption. With a further augment in the pH of the solution, the capacity of the sorbent decreased. This was due to the transition to the region of precipitation of metal hydroxides (pH ~ 7–10). Thus, the experimental data are in full agreement with the existing ideas about the course of exchange reactions on the surface of the nanocomposite. It should be noted that at pH = 9, the concentration in the solution (*C_e_*) is affected not only by the sorption activity of the sorbent but also by the formation of hydroxo complexes, which inevitably affects the final values of the adsorption capacity. Thus, further studies were conducted in buffer aqueous solutions for pH = 6.

#### 3.2.2. Dosage Effect

The study of dose-effect is also important in water treatment, especially on an economic and large scale. As a result of determining the effect of the adsorbent weight on its sorption capacity, under otherwise equal conditions (volume of solution, concentration, pH, etc.), it was found that the optimal weight of a sample of a nanocomposite for a solution volume of 30 mL was 0.005 g. Further increase in dose could not increase the removal capacity of the reported metal ions. Figure 8 presents the findings. This low amount of dose indicated a good sorption capacity of the nanocomposite material. Moreover, this low value of dose also dictates that the method is economic in nature.

#### 3.2.3. Time Effect

The effect of time on the removal of metal ions is very important for an economic adsorption method. Therefore, the optimization was carried out from 1.0 to 60 min time. The results are plated in Figure 9. A look at this Figure indicates that the sorption had increased from 1.0 to 15 min and then become constant. In this case, it is obvious that the curves have a linear section in the initial region, described by the Henry equation. This is followed by a gradual inflection and transition to a plateau in the saturation region. It is important to note that the sorbent demonstrated high sorption activity since more than 90% of the sorbate is extracted in the first 15 min of phase contact.

#### 3.2.4. Kinetics Study

The study of kinetics is very important to understanding the mechanism of sorption. To approximate the kinetic variables and define the possible mechanisms of sorption of heavy metal ions on the nanocomposite, the pseudo-first-order (Equation (4)) and pseudo-second-order (Equation (7)) models, the Elovich model (Equation (8)) and intraparticle diffusion (Equation (9)) were used. A model description of the sorption extraction of lead and zinc ions according to the above equations is shown in Figure 10.

The results of calculations of the key parameters of the kinetics of the process of adsorption of heavy metal ions by the nanocomposite are presented in Table 3 and Table 4.

According to the findings, the pseudo-second-order model (*R*^2^ = 0.99) demonstrates the best correlation, which indicates the chemical nature of the interaction between the metal ion and the active center of the nanocomposite. Also, a good approximation of the experimental data is shown by the intraparticle diffusion model (on average *R*^2^ = 0.98). We can assume the implementation of two processes occurs sequentially. At the same time, both of them occur quite efficiently, which led to high total values of the sorption capacity (and coefficients of determination). The first straight section can be attributed to diffusion in macropores (period 1), that is, to the transfer of contaminant molecules from the volume of the solution to the surface of the adsorbent, and the second linear section to diffusion in micropores (period 2), that is, heavy metal ions are sorbed on active centers. Thus, it can be argued that the adsorption processes on the nanocomposite under study have a mixed character.

#### 3.2.5. Isotherm Study

The construction of an adsorption isotherm made it probable to regulate the maximum adsorption capacity for the extracted component. The nature of the isotherm helps to interpret the sorption interactions between the active surface of the sorbent and the contaminants to be extracted and to establish the key features of sorption bonds. Figure 11 shows the sorption isotherms of lead and zinc ions on the synthesized cryogel.

For the theoretical interpretation of the adsorption isotherms of ions and molecules of various substances, a number of model dependencies are often used, such as Langmuir, Freundlich and Temkin. The linear forms of the graphs corresponding to the experimental isotherms of ion extraction are shown in Figure 12. The key properties of the sorption process can be understood in the presence of the conforming calculated parameters of the isotherm equations (Table 5).

According to Figure 12, the experimental sorption isotherms of lead and zinc ions are reliably described by the Langmuir equation, which indicated the occurrence of monomolecular sorption and the filling of the monolayer on the surface of the nanocomposite. The sorption capacity of the nanocomposite is weakly affected by the heterogeneity of its surface since the correlation of the experimental data (according to Temkin’s model) has a rather low *R^2^*. Using the Langmuir equation, the maximum theoretical adsorption values were obtained: *q_max_* (Pb) = 588 mg g^−1^, *q_max_* (Zn) = 400 mg g^−1^.

#### 3.2.6. The Influence of Temperature on the Sorption Capacity

The study of the thermodynamics of the adsorption process is important for understanding the nature of the ongoing interactions between the adsorbent and the contaminant. For this, experiments were carried out to determine the sorption capacity of the synthesized material using lead and zinc metal ions at different temperatures (Figure 13).

As can be seen from the presented dependences, with increasing temperature, the sorption activity of the prepared composite material increased. The highest adsorption capacity (830 mg g^−1^) was reached at a temperature of 45 °C. An important step is the correct choice of the constant for calculating the main thermodynamic parameters of the process. The method used in this work is described in detail in [54]. The plot of 1/T versus lnK (Figure 14); under the assumption that the enthalpy of reaction is a constant value in the narrow temperature range considered in this work; did not depend on the process temperature.

The presented graph with a good correlation fits into the assumption that the enthalpy is independent of temperature. Using the presented graphic dependence, it is possible to calculate the value of enthalpy, entropy and Gibbs free energy as given in Table 6.

Table 6 presents the main thermodynamic parameters of the sorption process. It is established that the enthalpy of the process is positive. This is typical for processes that increase efficiency with increasing temperature. The Gibbs free energy for this process is negative, which indicates the fundamental prospect of the spontaneous occurrence of the sorption procedure. The sorption entropy is positive. Perhaps this is due to an increase in the activity of ions in an aqueous solution associated with an increase in the process temperature. The enthalpy value of the sorption process is 36.9 $\mathrm{kJ}\cdot {\mathrm{mol}}^{-1}$. According to the well-known classification [36], this value is typical for transient heterogeneous processes of sorption extraction, in particular, ion exchange. This confirms the previously accepted expectations about the nature of the procedures under study. Thus, the conducted thermodynamic studies are in good agreement with the existing ideas about the nature of interactions between the nanocomposite and the extracted heavy metals.

#### 3.2.7. Desorption Study

To study the desorption by the synthesized nanocomposite, we chose the conditions for *q_max_*, which are achieved by extracting Zn and Pb ions at 45 °C.

The results of the experimental studies are presented in Figure 15.

The analysis of the experimental data shows that the highest desorption *R_des_* is ~0.275% for the Pb ions and ~0.583% for the Zn ions. The desorption curve naturally reaches a plateau, which indicates that a further increase in the concentration of ions in the sorbent did not lead to an increase in desorption. These low values of the desorption are a good indication of the strong adsorption of metal ions. It is due to the chemisorption nature of the metal ions removal.

Along with the desorption, an important indicator of component retention in the sorbent structure is the distribution constant calculated by the formula
$$K_{e}^{des}=\frac{(C_{0}-C_{e}-C_{e}^{des})V}{mC_{e}^{des}}=\frac{q_{e}^{des}}{\;C_{e}^{des}}$$

Using the presented calculation formulas and the experimental data obtained, the main parameters of the desorption process presented in Table 7 were assessed. The analysis of the presented data allows us to draw several conclusions. The value of the distribution constant indicates that the process of sorption extraction of heavy metal ions on the developed nanocomposite is practically irreversible.

### 3.3. Schematic Diagram of Zn (II) and Pb (II) Ions Adsorption on a Nanocomposite

The results of FT-IR Fourier and Raman spectroscopy confirmed the presence of reduced graphene oxide, oxidized carbon nanotubes, polyaniline and phenol-formaldehyde resin in the nanocomposite. Zinc and lead ions (which are designated in the scheme of Figure 16 as Me^2+^) are Lewis acids and have the ability to form coordination bonds with the main groups, which can be the alcohol groups of the phenol-formaldehyde resin, the amino groups of polyaniline and the oxygen-containing groups (phenolic and carboxyl); present on the surface of carbon nanostructures [37]. Figure 16 shows the bonds formed between Zn (II) and Pb (II) ions and fragments of the PANI molecule. In addition, Figure 16 also shows two possible options for the formation of coordination bonds with fragments of phenol-formaldehyde resin molecules. Oxygen-containing groups may partially remain on the surface of r-GO and o-CNTs. Figure 16 also shows options for the formation of bonds between metal ions and oxygen-containing groups on the surface of CNTs and r-GO. Thus, each component of the developed composite has the ability to bind Me^2+^ ions, which together leads to a high sorption activity of the resulting functional nanomaterial.

## 4. Conclusions

The article proposed a new nanocomposite material based on reduced graphene oxide and oxidized CNTs modified with polyaniline. To stabilize the composite in an aqueous medium, the phenol-formaldehyde resin was added as a component. Physical-chemical, textural, structural and morphological properties were studied using SEM and TEM, Raman and IR-Fourier spectroscopy, XRD and TG/DSC. To preserve the structure and volume of the porous space at the final stage of preparation, the nanocomposite was dried using lyophilic treatment. In this study, the sorption properties of nanocomposite were studied for the elimination of lead and zinc metal ions from water. The adsorption kinetics of metal ions removal was pseudo-second-order, Elovich and intraparticle diffusion models. When describing the equilibrium state of adsorption, the Langmuir and Temkin models were fitted well. The removal capacities were 346 and 581 mg/g at pH 6.5, 0.005 g/30 mL dose and 15 min contact time. These conditions made this method highly applicable, nature-friendly and fast. Therefore, the reported method may be used successfully for the removal of lead and zinc from water.

## Figures and Tables

**Figure 1 molecules-27-05623-f001:**
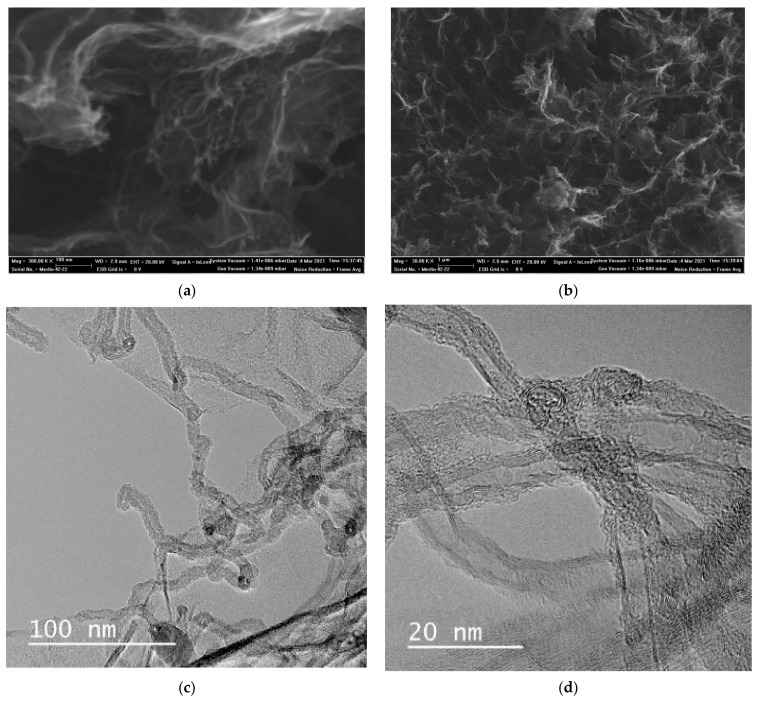
SEM (**a**,**b**) and TEM (**c**,**d**) image of the surface of the nanocomposite cryogel.

**Figure 2 molecules-27-05623-f002:**
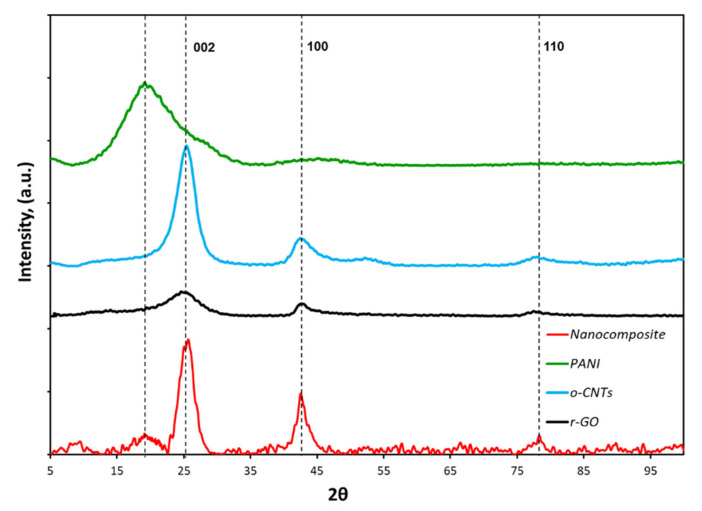
XRD-pattern of the initial materials and the r-GO/o-CNTs/PANI nanocomposite cryogel.

**Figure 3 molecules-27-05623-f003:**
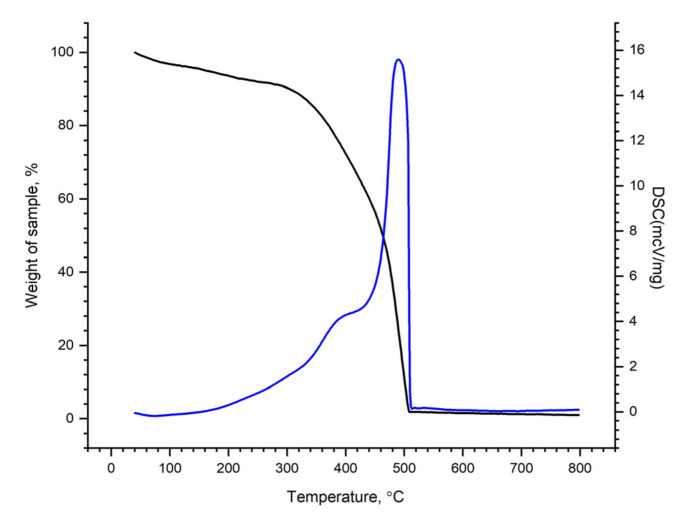
TG-DSC-curve of the r-GO/o-CNTs/PANI nanocomposite cryogel.

**Figure 4 molecules-27-05623-f004:**
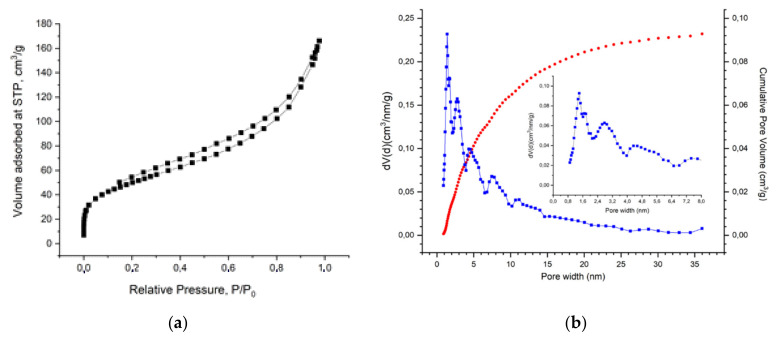
Pore space parameters of the r-GO/o-CNTs/PANI nanocomposite cryogel: (**a**) nitrogen adsorption-desorption isotherms, (**b**) pore size distribution obtained from DFT calculations assuming spherical/cylindrical pores (QSDFT Model).

**Figure 5 molecules-27-05623-f005:**
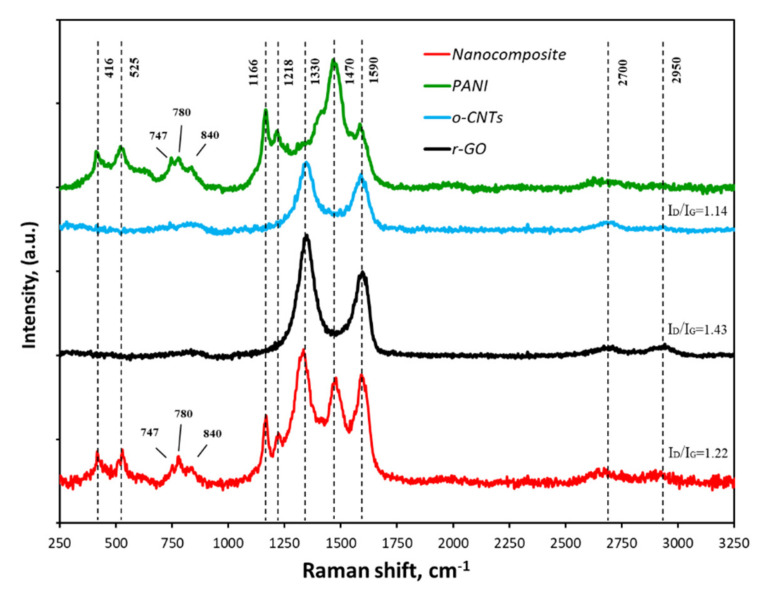
Raman-spectra of the initial materials and nanocomposite.

**Figure 6 molecules-27-05623-f006:**
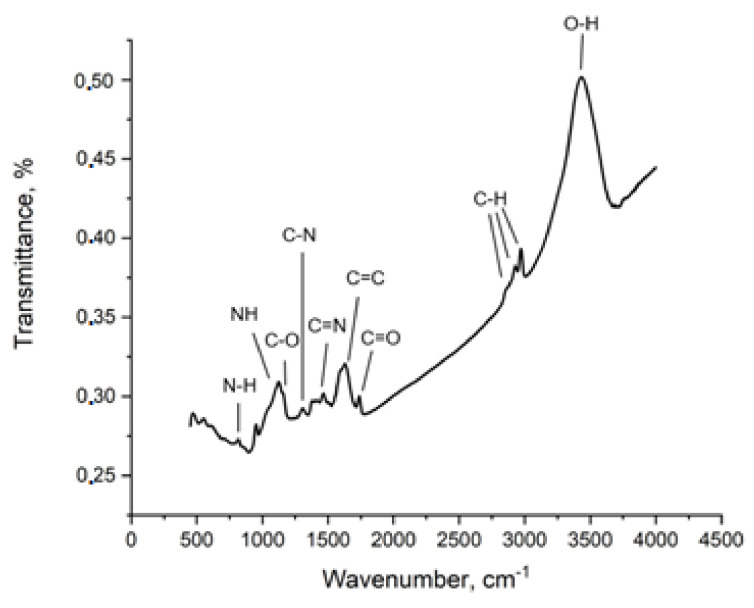
IR-spectrum of the r-GO/o-CNTs/PANI nanocomposite cryogel.

**Figure 7 molecules-27-05623-f007:**
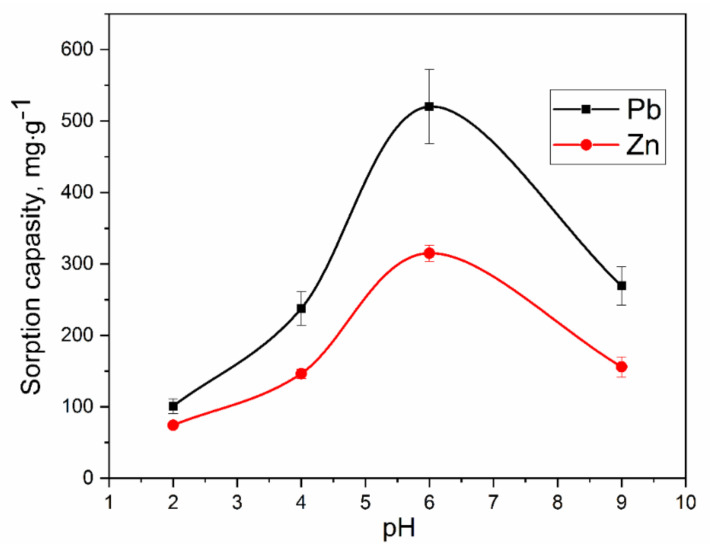
The influence of pH of aqueous solution on the sorption of lead and zinc ions in buffer systems.

**Figure 8 molecules-27-05623-f008:**
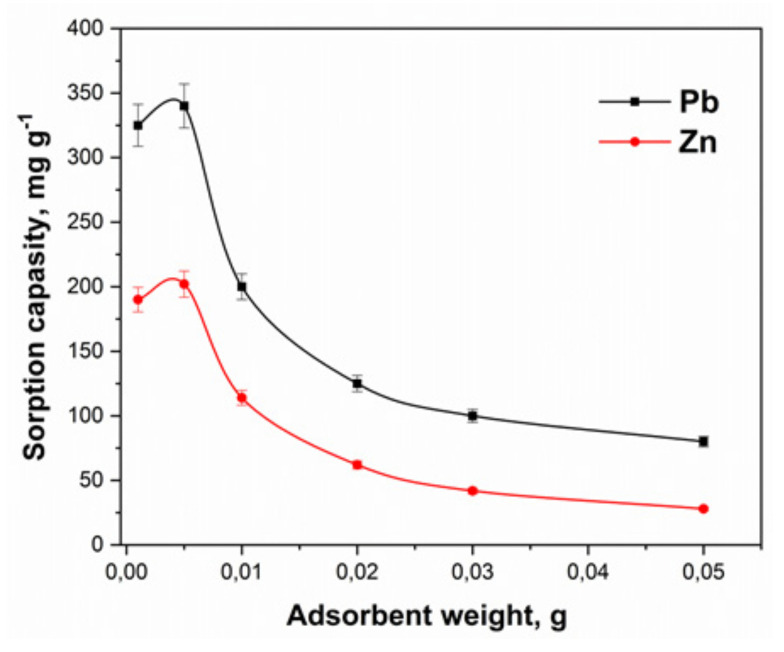
The influence of sorbent weight on sorption of lead and zinc ions in acetic-acetate buffer systems.

**Figure 9 molecules-27-05623-f009:**
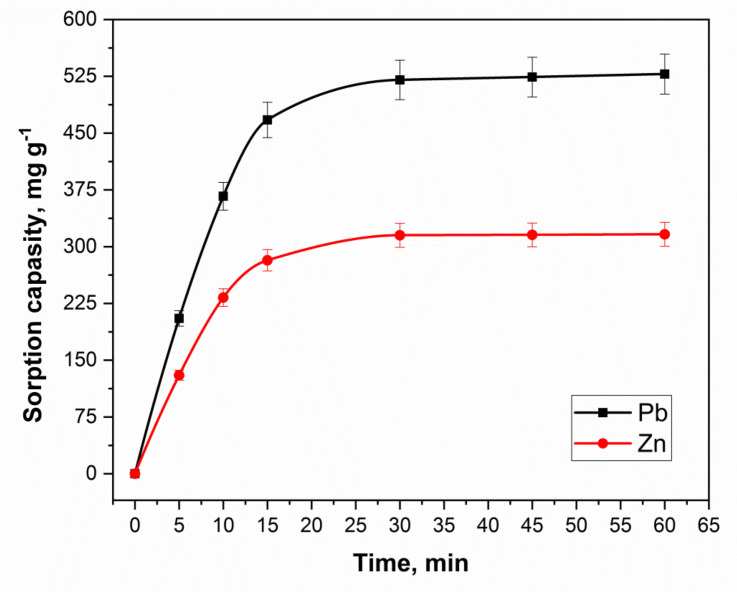
Kinetics of the Pb^2+^ and Zn^2+^ sorption onto nanocomposite.

**Figure 10 molecules-27-05623-f010:**
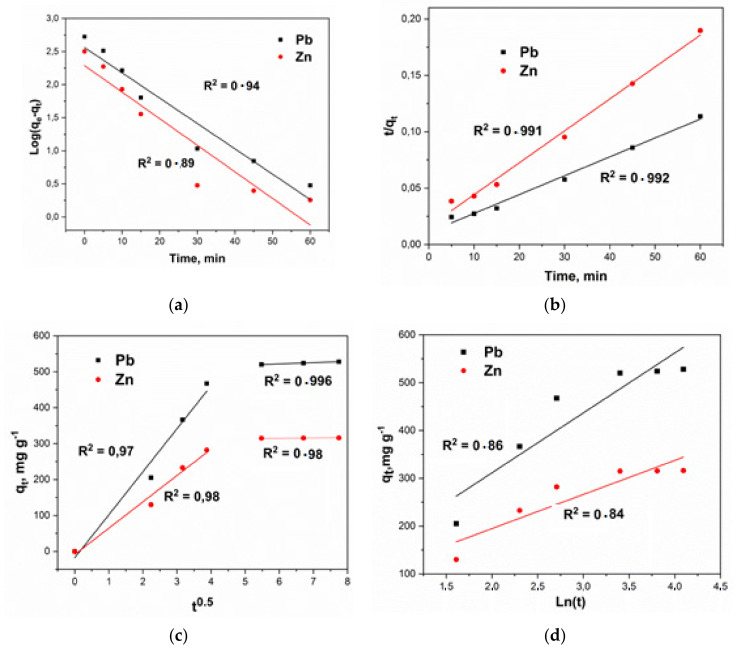
Kinetic modeling: (**a**) Pseudo-first order, (**b**) Pseudo-second order, (**c**) Intraparticle diffusion model, (**d**) Elovich model.

**Figure 11 molecules-27-05623-f011:**
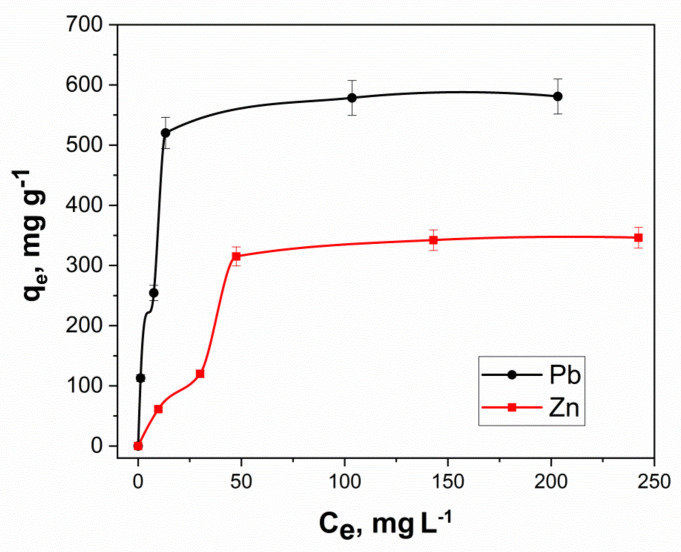
The isotherms of Pb^2+^ and Zn^2+^ sorption onto nanocomposite.

**Figure 12 molecules-27-05623-f012:**
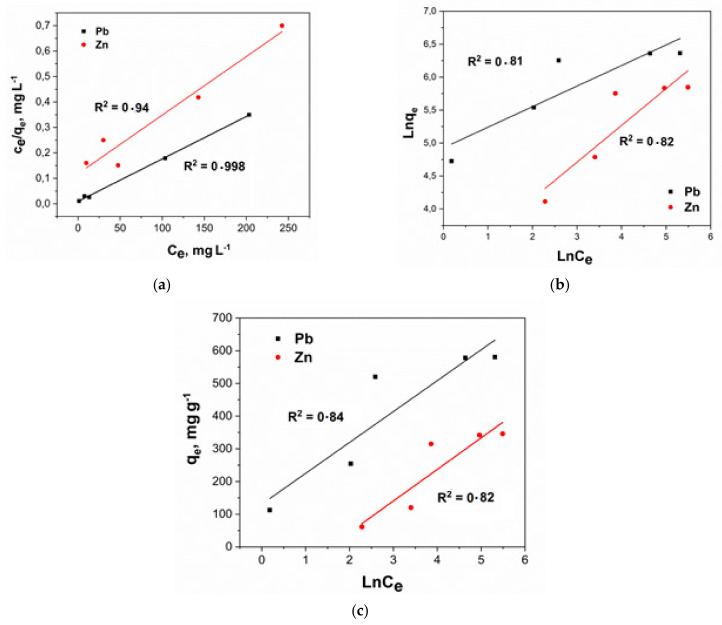
Linear forms of the Langmuir (**a**), Freindlich (**b**) and Temkin (**c**) equations.

**Figure 13 molecules-27-05623-f013:**
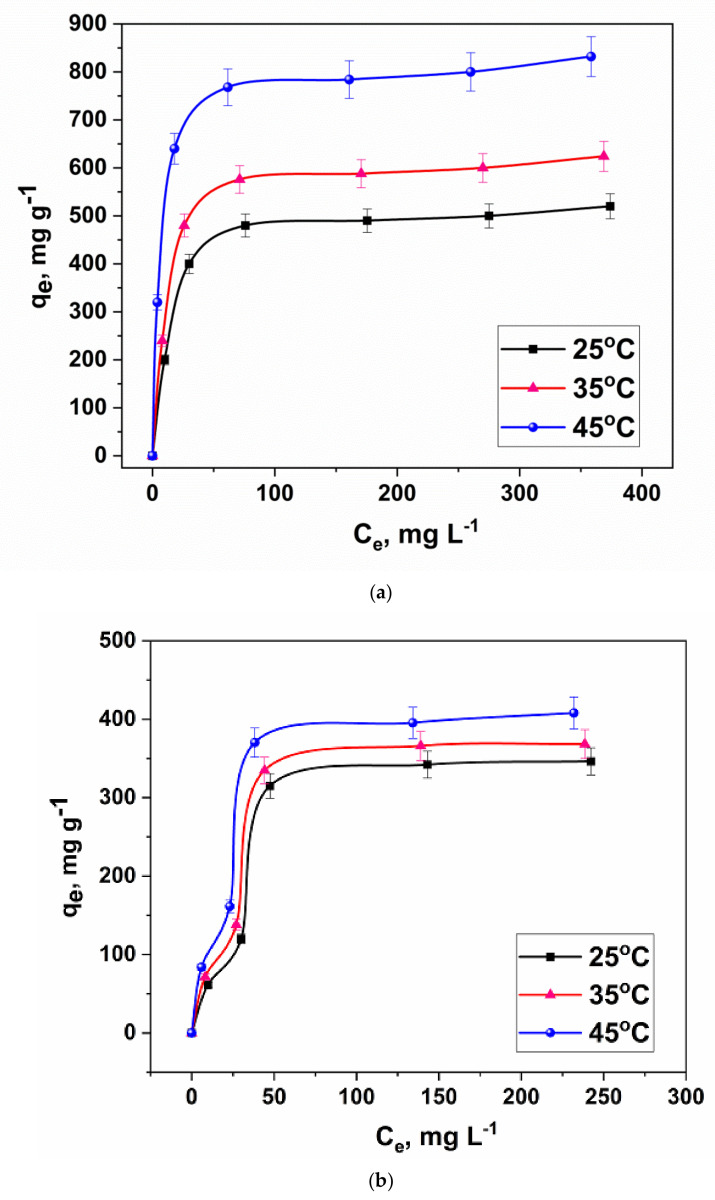
Sorption isotherms of (**a**): lead and (**b**): zinc ions by a nanocomposite at different temperatures.

**Figure 14 molecules-27-05623-f014:**
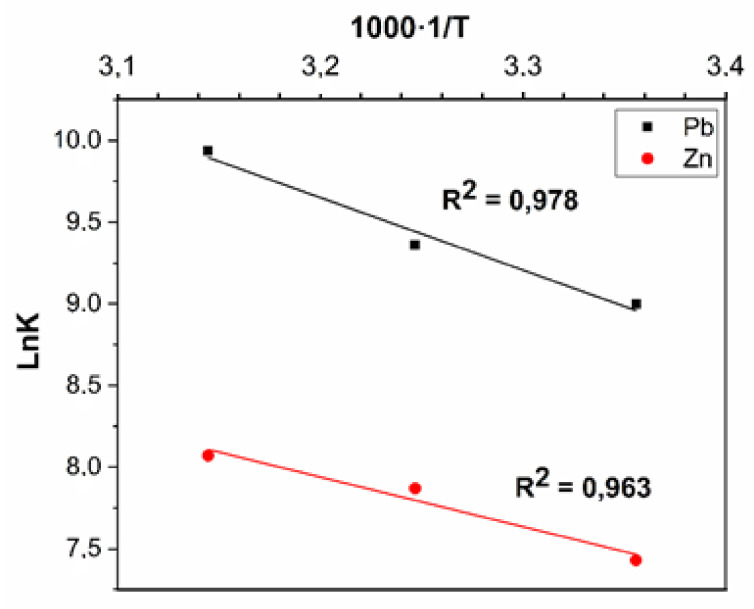
Dependence graph of lnK versus reciprocal temperature.

**Figure 15 molecules-27-05623-f015:**
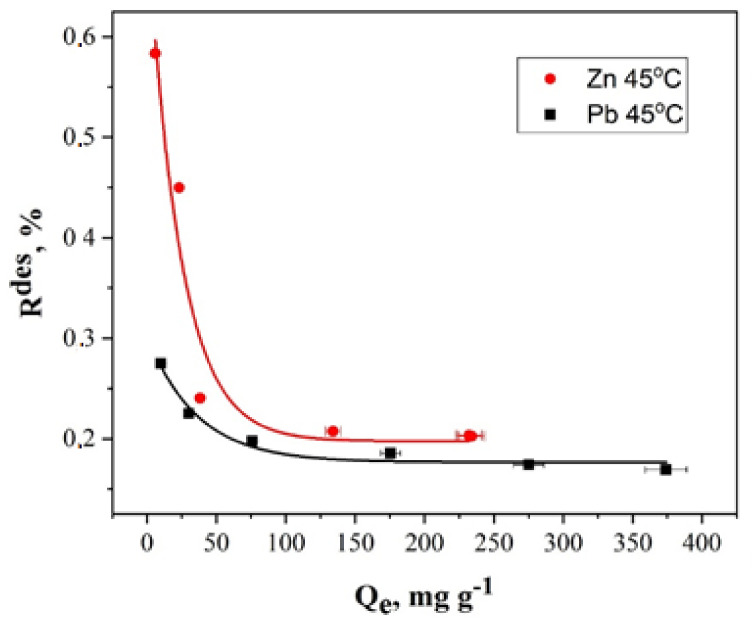
Dependence of the desorption on the nanocomposite adsorption capacity toward the Zn and Pb ions.

**Figure 16 molecules-27-05623-f016:**
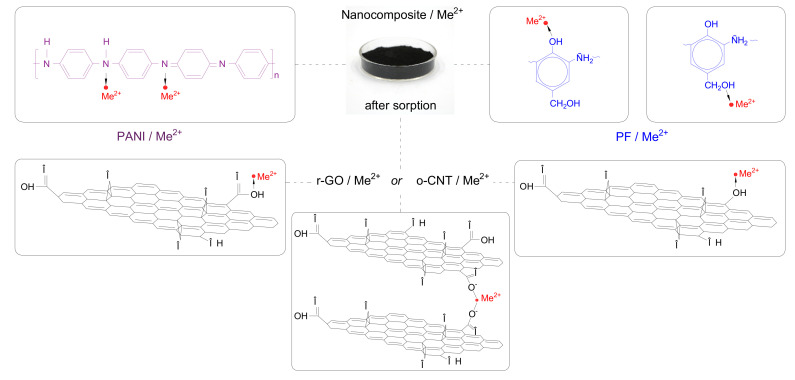
The proposed mechanism for the adsorption of contaminants on the nanocomposite.

**Table 1 molecules-27-05623-t001:** Interplanar distances of initial materials.

Name	Interplanar Distance (d) for Angle 2θ in the Region of 19°, Å	Interplanar Distance (d) for Angle 2θ in the Region of 25°, Å	Interplanar Distance (d) for Angle 2θ in the Region of 42°, Å	Interplanar Distance (d) for Angle 2θ in the Region of 77°, Å
CNTs		3.50 (25.41°)	2.13 (42.40°)	1.23 (77.81°)
polyaniline	4.64 (19.11°)		2.12 (42.63°)	
r-GO		3.58 (24.88°)	2.15 (42.10°)	1.23 (77.24°)

**Table 2 molecules-27-05623-t002:** Interplanar distances for the final nanocomposite.

Material	Interplanar Distance (d) for Angle 2θ in the Region 19°, Å	Interplanar Distance (d) for Angle 2θ in the Region 25°, Å	Interplanar Distance (d) for Angle 2θ in the Region 42°, Å	Interplanar Distance (d) for Angle 2θ in the Region 77°, Å
Cryogel	4.641 (19.11°)	3.476 (25.60°)	2.120 (42.61°)	1.215 (78.68°)

**Table 3 molecules-27-05623-t003:** Parameters of the sorption kinetics of lead ions.

Pseudo-first order (a)				Pseudo-second order (b)		
*q_e_*	*k* _1_		*R* ^2^	*q_e_*	*k* _2_	*R* ^2^
364	0.09		0.94	588	0.00027	0.991
Intraparticle diffusion model (c):				Elovich model (d)		
	*k_id_*	*C*	*R* ^2^	*α*	*β*	*R* ^2^
Period 1	119.8	17.9	0.977	3.5 × 10^2^	3.46 × 10^−3^	0.860
Period 2	3.43	501.3	0.996			

**Table 4 molecules-27-05623-t004:** Parameters of the sorption kinetics of zinc ions.

Pseudo-first order (a)				Pseudo-second order(b)		
*q_e_*	*k* _1_		*R* ^2^	*q_e_*	*k* _2_	*R* ^2^
194	0.092		0.89	357	0.0005	0.992
Intraparticle diffusion model (c):				Elovich model (d)		
	*k_id_*	*C*	*R* ^2^	*α*	*β*	*R* ^2^
Period 1	73.4	8.83	0.98	1.46 × 10^2^	1.4 × 10^−2^	0.84
Period 2	0.52	312	0.98			

**Table 5 molecules-27-05623-t005:** Isotherms parameters for lead and zinc sorption on the nanocomposite.

Langmuir			Freundlich			Temkin		
Pb								
*q_max_*	*K_L_*	*R* ^2^	*b_F_*	*K_F_*	*R* ^2^	*B_T_*	*K_T_*	*R* ^2^
588	0.25	0.998	3.2	8.51 × 10^3^	0.81	94.5	3.9	0.837
Zn								
*q_max_*	*K_L_*	*R* ^2^	*b_F_*	*K_F_*	*R* ^2^	*B_T_*	*K_T_*	*R* ^2^
400	0.03	0.94	1.79	1.07 × 10^3^	0.82	96.5	4.6	0.82

**Table 6 molecules-27-05623-t006:** Thermodynamic parameters of lead and zinc ions in the adsorption process.

T, K	1/T	Ln(K)		ΔH $kJ\cdot mol^{-1}$		ΔG $kJ\cdot mol^{-1}$		ΔS $J\cdot mol^{-1}\cdot K$	
		Pb	Zn	Pb	Zn	Pb	Zn	Pb	Zn
298	0.0033	8.99	7.43	36.9	26.6	−22.3	−4.97	158.6	149.9
308	0.0032	9.36	7.87			−23.9	−5.28		
318	0.0031	9.94	8.07			−26.3	−5.52		

**Table 7 molecules-27-05623-t007:** Calculated parameters of the desorption process for both ions.

Pb				Zn			
$C_{e\;}$, mg L^−1^	$C_{e}^{des}$, mg L^−1^	$K_{e}^{des}\cdot 10^{-3}$	$R_{des},\;\%$	$C_{e\;},\;\mathrm{mg}\;L{-}^{1}$	$C_{e}^{des},\;\mathrm{mg}\;L{-}^{1}$	$K_{e}^{des}\cdot 10^{-2}$	$R_{des},\;\%$
10	0.55	1.04	0.275	6.0	0.49	1.71	0.583
30	0.9	3.44	0.225	23.1	0.85	1.90	0.451
76	0.95	4.38	0.198	38.3	0.89	4.16	0.240
175,5	0.91	4.29	0.185	134.1	0.82	4.82	0.207
275	0.87	4.27	0.174	234	0.81	4.93	0.202
374	0.88	4.42	0.169	232	0.83	4.91	0.203

## Data Availability

All data is given here.

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
