# Peer review of "Polyaniline Modified CNTs and Graphene Nanocomposite for Removal of Lead and Zinc Metal Ions: Kinetics, Thermodynamics and Desorption Studies"

_molecules, 2022, doi:10.3390/molecules27175623_

Round 1

Reviewer 1 Report

This manuscript designed a nanocomposite based on reduced graphene oxide, polyaniline and carbon oxide nanotube for effective removal of the lead and zinc ions in water. The structural information, adsorption kinetics, isotherms, thermodynamics and desorption properties of this nanocomposite were detailed studied. However, in my opinion it should not be accepted for publication before major revision. All recommendations are briefly presented below:

1.     What does the size 2.32-7.34 nm (Page 1, line 17) mean? It is obvious not the length of graphene oxide and carbon nanotubes.

2.     This manuscript reported effective removal of the lead and zinc metal ions by polyaniline modified CNTs and graphene based nanocomposites. Contrast experiments should be given, such as the removal ability of the lead and zinc metal ions by polyaniline modified CNTs and polyaniline modified graphene.

3.     The authors mentioned the reference that the graphene oxide showed excellent adsorption capacity of 1119 mg g-1 for Pb (II) ions at pH = 5 (Page 2, line 70-71), which is much higher than the number (581 mg/g at pH 6.5) in this manuscript. 

4.     For better comparison, the IR-spectrum of PANI needs to be provided.

5.     Some revision remarks were kept in the manuscript. (Page 30 line 868, Page 28 line 788)

6.     There are many obvious minor mistakes. Authors should check carefully throughout the manuscript. The following are only several examples.

1) Page 1 line 18, “176 m2/g” should be corrected to “176 m2/g”

2)     Page 2 line 56,2.6; 5.1” should be corrected to “2.6, 5.1”.

3)     Page 3 line 98, the typeface of “heavy metal ions” is different from other words.

4)     Page 11 line 400, the degree expression is not correct.

5)     The references are not given in the same format.

Author Response

Pointwise replies

Manuscript ID: molecules- 1855563

Title: Polyaniline modified CNTs and graphene nanocomposite for removal of lead and zinc metal ions: Kinetics, thermodynamics and desorption studies.

First of all, I would like to thank Ms. Lsla Li, Section Managing Editor, to give us a chance for revising this manuscript. Besides, thanks are also the scholarly reviewer to give fruitful suggestions. Really, the incorporation of all the suggestions made this manuscript more useful and attractive to the readers. The point-wise replies to the comments of reviewers are given below.

Open Review

English language and style

( ) Extensive editing of English language and style required
( ) Moderate English changes required
(x) English language and style are fine/minor spell check required
( ) I don't feel qualified to judge about the English language and style

Yes

Can be improved

Must be improved

Not applicable

Does the introduction provide sufficient background and include all relevant references?

( )

(x)

( )

( )

Are all the cited references relevant to the research?

(x)

( )

( )

( )

Is the research design appropriate?

( )

( )

(x)

( )

Are the methods adequately described?

(x)

( )

( )

( )

Are the results clearly presented?

(x)

( )

( )

( )

Are the conclusions supported by the results?

(x)

( )

( )

( )

Comments and Suggestions for Authors

This manuscript designed a nanocomposite based on reduced graphene oxide, polyaniline and carbon oxide nanotube for effective removal of the lead and zinc ions in water. The structural information, adsorption kinetics, isotherms, thermodynamics and desorption properties of this nanocomposite were detailed studied. However, in my opinion it should not be accepted for publication before major revision. All recommendations are briefly presented below:

Reply:

Thanks to this reviewer for his/her appreciation of our work.

Also, thanks for sparing his/her valuable time reviewing this manuscript and giving fruitful suggestions.

  1. What does the size 2.32-7.34 nm (Page 1, line 17) mean? It is obvious not the length of graphene oxide and carbon nanotubes.

Reply:

It is the size of the synthesized nanocomposite.

  1. This manuscript reported effective removal of the lead and zinc metal ions by polyaniline modified CNTs and graphene based nanocomposites. Contrast experiments should be given, such as the removal ability of the lead and zinc metal ions by polyaniline modified CNTs and polyaniline modified graphene.

Reply:

We did experiments with the removal ability of the lead and zinc metal ions by polyaniline modified CNTs and polyaniline modified graphene BUT the results were not good and discarded.

  1. The authors mentioned the reference that the graphene oxide showed excellent adsorption capacity of 1119 mg g-1for Pb (II) ions at pH = 5 (Page 2, line 70-71), which is much higher than the number (581 mg/g at pH 6.5) in this manuscript. 

Reply:

It was typing mistake and rectified.

  1. For better comparison, the IR-spectrum of PANI needs to be provided.

Reply:

It is given in Figure 6.

  1. Some revision remarks were kept in the manuscript. (Page 30 line 868, Page 28 line 788)

Reply:

These are taken care.

  1. There are many obvious minor mistakes. Authors should check carefully throughout the manuscript. The following are only several examples.

Reply:

These are taken care.

1) Page 1 line 18, “176 m2/g” should be corrected to “176 m2/g”

Reply:

It is corrected as suggested.

2)     Page 2 line 56, “2.6; 5.1” should be corrected to ““2.6, 5.1”.

Reply:

It is corrected as suggested.

3)     Page 3 line 98, the typeface of “heavy metal ions” is different from other words.

Reply:

It is corrected as suggested.

4)     Page 11 line 400, the degree expression is not correct.

Reply:                                                                

It is corrected as suggested.

5)     The references are not given in the same format.

Reply:

The references are formatted according to the Journal.

ONCE AGAIN WE ARE HIGH THANKFUL TO THIS REVIEWER FOR SPARING HIS/HER VALUABLE TIME AND GIVING US THE FRUITFUL SUGGESTIONS.

Reviewer 2 Report

The present study aims to synthesize a hybrid nanostructured adsorbent based on r-GO modified with PANI. In this method, o-CNTs played a role of a structure-forming agent and as an additional component that improved the efficiency of zinc and leads metal ions removal. The methods are interesting but the authors didn’t discuss them very well. It is suggested to consider the following issue:

-the Title is too lengthy and should be rewritten in a summarized form

-what is the novelty of this study, there are many similar papers with these NPs.

-the removal of contaminant from water by different micro and nanostructures were reported in many papers. it suggested discussing some other structures that are used for water removal and compare with yours: https://doi.org/10.1016/j.micromeso.2021.111289, https://doi.org/10.1016/j.jhazmat.2021.128121, https://doi.org/10.1016/j.jece.2021.107091, https://doi.org/10.1016/j.cej.2021.133166

-there are many typo errors in the manuscript. For example, line 168: 2.3.С. characterization methods

-how samples were prepared for SEM and TEM? The detailed methodology was expected

-In figure 1 a and b, the error bar didn’t observe clearly, please add a clear error bar

-line 386-388 please add related references for your claim

-please add a graphical abstract to present the goal of the current study

Author Response

Pointwise replies

Manuscript ID: molecules- 1855563

Title: Polyaniline modified CNTs and graphene nanocomposite for removal of lead and zinc metal ions: Kinetics, thermodynamics and desorption studies.

First of all, I would like to thank Ms. Lsla Li, Section Managing Editor, to give us a chance for revising this manuscript. Besides, thanks are also the scholarly reviewer to give fruitful suggestions. Really, the incorporation of all the suggestions made this manuscript more useful and attractive to the readers. The point-wise replies to the comments of reviewers are given below.

Open Review

English language and style

(x) Extensive editing of English language and style required
( ) Moderate English changes required
( ) English language and style are fine/minor spell check required
( ) I don't feel qualified to judge about the English language and style

Yes

Can be improved

Must be improved

Not applicable

Does the introduction provide sufficient background and include all relevant references?

( )

( )

(x)

( )

Are all the cited references relevant to the research?

( )

( )

(x)

( )

Is the research design appropriate?

( )

( )

(x)

( )

Are the methods adequately described?

( )

( )

(x)

( )

Are the results clearly presented?

( )

( )

(x)

( )

Are the conclusions supported by the results?

( )

( )

(x)

( )

Comments and Suggestions for Authors

The present study aims to synthesize a hybrid nanostructured adsorbent based on r-GO modified with PANI. In this method, o-CNTs played a role of a structure-forming agent and as an additional component that improved the efficiency of zinc and leads metal ions removal. The methods are interesting but the authors didn’t discuss them very well. It is suggested to consider the following issue:

Reply:                                      

Thanks to this reviewer for his/her appreciation of our work.

Also, thanks for sparing his/her valuable time reviewing this manuscript and giving fruitful suggestions.

-the Title is too lengthy and should be rewritten in a summarized form

Reply:

The title is condensed as given below.

Polyaniline modified CNTs and graphene nanocomposite for removal of lead and zinc metal ions: Kinetics, thermodynamics and desorption studies

-what is the novelty of this study, there are many similar papers with these NPs.

Reply:

This is the first paper describing nanocomposite of polyaniline, CNTs and graphene.

The novelty of the work lies in the development of a method for the synthesis of a new sorption material based on special carbon nanostructures - reduced graphene oxide and oxidized CNTs (as universal sorbents), which are further improved by polyaniline - an organic polymer that significantly increases the overall efficiency of the final product. The obtained material was synthesized in a cryogel form, which having an easily accessible highly porous structure with a large number of active sorption sites. The chemical composition of this material and the technology of its production represent the main scientific importance of the work.

-the removal of contaminant from water by different micro and nanostructures were reported in many papers. it suggested discussing some other structures that are used for water removal and compare with yours: https://doi.org/10.1016/j.micromeso.2021.111289, https://doi.org/10.1016/j.jhazmat.2021.128121, https://doi.org/10.1016/j.jece.2021.107091, https://doi.org/10.1016/j.cej.2021.133166

Reply:

The suggested 3 papers belong to the removal of dyes and phosphate and the 4th one is for photo-degradation. The results of these papers cannot be matched with ours as our paper deals with the removal of metal ions. However, to ACCOMMODATE TWO PAPERS we added a few lines of adsorption and materials and cite these papers.

-there are many typo errors in the manuscript. For example, line 168: 2.3.С. characterization methods

Reply:

These are corrected.

-how samples were prepared for SEM and TEM? The detailed methodology was expected

Reply:

The samples are prepared as per the standard procedure. These procedures are normal and available everywhere. We did not include in this manuscript as the manuscript is already large.

-In figure 1 a and b, the error bar didn’t observe clearly, please add a clear error bar

Reply:

Figures 1 a and b are of SEM and TEM and it is not possible to include error bars. Such figures never have error bars.

-line 386-388 please add related references for your claim

Reply:

It is taken care of.

 -please add a graphical abstract to present the goal of the current study

Reply:

Graphical abstract is uploaded separately on Journal home page.

ONCE AGAIN WE ARE HIGH THANKFUL TO THIS REVIEWER FOR SPARING HIS/HER VALUABLE TIME AND GIVING US THE FRUITFUL SUGGESTIONS.

Reviewer 3 Report

In this manuscript, the author focused on the synthesis of a hybrid nanostructured adsorbent based on r-GO modified with Polyaniline, for the removal of metal ions. There is no novelty in your work and so many reports are available using GO with polyaniline. Hence, I recommended a major review and it is accepted for this journal after the author clarifies the following comments.

1- This manuscript has several typos (e.g. line 18/19: units)

2- Introduction; many statements lack appropriate references. In general, referencing has not been taken into account in many parts such as introduction and discussions.

3- The authors may need to briefly address the difference(s) between the current manuscript and other similar published review articles in the Introduction section.

5- Please explain why you have used polyaniline as modifying agent over other compounds, for example biopolymers or other bio molecules. What is the advantage?

6- Lines 192-193: In specific surface area analysis (BET), which pressure and temperature were used for degassing procedure? Which P/P0 values were used in the calculation of specific surface areas? 

7- Lines 197-199: Regarding to the FTIR measurements, it should be important to mention the scans used and also the resolution. Did the authors performed the FTIR measurements using the ATR mode or using KBr pellets?

8- Sections 2.5.1 and 2.5.2: Please do not use linearization of the equations. Nowadays, most computer programs can perform non-linear regression and should be used in preference of linearization to determine adsorption parameters.

9- The authors should make comparison with literature for the metal ions used.

10- Which type of water was used to prepare the solutions with de dyes: ultra-pure water, distilled water or deionized water? This detail is important for the readers.

11- Conclusions need to be improved by specifying the discussed important points within this work. In the conclusions, the authors should also provide an outlook of the challenges and potential future directions.

Other comments:

Overall, the materials used should be more characterized. For instance, I suggest the authors to measure zeta potential values of the materials as function of pH and to measure the specific surface area.

The biggest concern with the approach presented here is the potential for sorption of unintended species in the water. Reasonably capacity is demonstrated, which is important, but selectivity is at least as important as capacity. The authors show that the sorbent can capture the metal ions, but this study does not evaluate how many interfering species are also captured, which will limit the practical effectiveness. This topic should be addressed explicitly in the discussion.

Could you comment on whether there is aging of the samples during time? This is absolutely important for practical applications.

In order to make the adsorption process more feasible, the adsorbent is usually regenerated. Data regarding the recycling performance of the material should be added in the manuscript. Moreover, please provide the XRD and TEM, results of the adsorbent after the five times of regeneration.

Author Response

Pointwise replies

Manuscript ID: molecules- 1855563

Title: Polyaniline modified CNTs and graphene nanocomposite for removal of lead and zinc metal ions: Kinetics, thermodynamics and desorption studies.

First of all, I would like to thank Ms. Lsla Li, Section Managing Editor, to give us a chance for revising this manuscript. Besides, thanks are also the scholarly reviewer to give fruitful suggestions. Really, the incorporation of all the suggestions made this manuscript more useful and attractive to the readers. The point-wise replies to the comments of reviewers are given below.

Open Review

English language and style

( ) Extensive editing of English language and style required
( ) Moderate English changes required
( ) English language and style are fine/minor spell check required
(x) I don't feel qualified to judge about the English language and style

Yes

Can be improved

Must be improved

Not applicable

Does the introduction provide sufficient background and include all relevant references?

( )

( )

(x)

( )

Are all the cited references relevant to the research?

( )

( )

(x)

( )

Is the research design appropriate?

( )

( )

(x)

( )

Are the methods adequately described?

( )

( )

(x)

( )

Are the results clearly presented?

( )

( )

(x)

( )

Are the conclusions supported by the results?

( )

( )

(x)

( )

Comments and Suggestions for Authors

In this manuscript, the author focused on the synthesis of a hybrid nanostructured adsorbent based on r-GO modified with Polyaniline, for the removal of metal ions. There is no novelty in your work and so many reports are available using GO with polyaniline. Hence, I recommended a major review and it is accepted for this journal after the author clarifies the following comments.

Reply:

Thanks to this reviewer for his/her appreciation of our work.

Also, thanks for sparing his/her valuable time reviewing this manuscript and giving fruitful suggestions.

1- This manuscript has several typos (e.g. line 18/19: units)

Reply:

The typos have been rectified.

2- Introduction; many statements lack appropriate references. In general, referencing has not been taken into account in many parts such as introduction and discussions.

Reply:

            We have added the references wherever are required.

3- The authors may need to briefly address the difference(s) between the current manuscript and other similar published review articles in the Introduction section.

Reply:

We have already cited the literature in the introduction.

 5- Please explain why you have used polyaniline as modifying agent over other compounds, for example biopolymers or other bio molecules. What is the advantage?

Reply:

Polyaniline (PANI) is a famous conductive polymer, and it has received tremendous consideration from researchers in the field of nanotechnology for the improvement of material properties including sorbents. PANI is doped easily because of its easy synthesis and remarkable environmental stability. PANI is one of the most effective organic polymeric materials used to remove various cationic and anionic pollutants from aqueous media, due to the significant content of phenylenediamine and quinonediamine functional groups. Due to its high nitrogen functionality, this polymer is often used in the extraction of heavy metal ions from aqueous media.

6- Lines 192-193: In specific surface area analysis (BET), which pressure and temperature were used for degassing procedure? Which P/P0 values were used in the calculation of specific surface areas? 

Reply:

The degassing procedure was carried out at 200°Ð¡ and a pressure of 1.3·10-3 Pa for 8-10 hours. For calculating the specific surface area, P/P0 values up to 0.3 were used.

7- Lines 197-199: Regarding to the FTIR measurements, it should be important to mention the scans used and also the resolution. Did the authors performed the FTIR measurements using the ATR mode or using KBr pellets?

Reply:

FTIR measurements were performed using KBr pellets. Number of scans - 16. Resolution - 4 cm-1.

8- Sections 2.5.1 and 2.5.2: Please do not use linearization of the equations. Nowadays, most computer programs can perform non-linear regression and should be used in preference of linearization to determine adsorption parameters.

Reply:

            We tried both linear and non-linear equations but the data are described well by linear equations.

9-  The authors should make comparison with literature for the metal ions used.

Reply:

            This is already given in te introduction part.

10- Which type of water was used to prepare the solutions with de dyes: ultra-pure water, distilled water or deionized water? This detail is important for the readers.

 Reply:

This is not the dye manuscript BUT is dealing metal ions solution. The solutons were prepared in deionized water.

11- Conclusions need to be improved by specifying the discussed important points within this work. In the conclusions, the authors should also provide an outlook of the challenges and potential future directions.

Reply:

This is provided in the conclusion part.

Other comments:

  • Overall, the materials used should be more characterized. For instance, I suggest the authors to measure zeta potential values of the materials as function of pH and to measure the specific surface area.

Reply:

            The surface area was analyzed with BET method. We have only this facility in our laboratory.

  • The biggest concern with the approach presented here is the potential for sorption of unintended species in the water. Reasonably capacity is demonstrated, which is important, but selectivity is at least as important as capacity. The authors show that the sorbent can capture the metal ions, but this study does not evaluate how many interfering species are also captured, which will limit the practical effectiveness. This topic should be addressed explicitly in the discussion.

Reply:

            After developing the method, it was applied in the river water and river water has many constituents under such circumstances the sorbet has sorption cappcities of 346, and 581 mg/g for zinc and lead.

  • Could you comment on whether there is aging of the samples during time? This is absolutely important for practical applications.

Reply:

We prepared samples, washed, dried and stored them. After these treatments samples are supposed to be sable. However, we tested the samples just after their preparation and after 3 months. There was no change in the results; indicating the stability of the samples.

kept the samples for 3 months before using

ONCE AGAIN WE ARE HIGH THANKFUL TO THIS REVIEWER FOR SPARING HIS/HER VALUABLE TIME AND GIVING US THE FRUITFUL SUGGESTIONS.

Round 2

Reviewer 1 Report

This manuscript improves a lot. Now the manuscript could be accepted.Text editing should be double checked (e.g.Page 1 line 61, “0.232 cm3/g” should be corrected to “0.232 cm3/g”).

Author Response

Pointwise replies

Manuscript ID: molecules- 1855563

Title: Polyaniline modified CNTs and graphene nanocomposite for removal of lead and zinc metal ions: Kinetics, thermodynamics and desorption studies.

First of all, I would like to thank Ms. Jacky Zhang, Section Managing Editor, to give us a chance for revising this manuscript. Besides, thanks are also the scholarly reviewer to give fruitful suggestions. Really, the incorporation of all the suggestions made this manuscript more useful and attractive to the readers. The point-wise replies to the comments of reviewers are given below.

Open Review

English language and style

( ) Extensive editing of English language and style required
( ) Moderate English changes required
(x) English language and style are fine/minor spell check required
( ) I don't feel qualified to judge about the English language and style

Yes

Can be improved

Must be improved

Not applicable

Does the introduction provide sufficient background and include all relevant references?

(x)

( )

( )

( )

Are all the cited references relevant to the research?

(x)

( )

( )

( )

Is the research design appropriate?

(x)

( )

( )

( )

Are the methods adequately described?

(x)

( )

( )

( )

Are the results clearly presented?

(x)

( )

( )

( )

Are the conclusions supported by the results?

(x)

( )

( )

( )

Comments and Suggestions for Authors

This manuscript improves a lot. Now the manuscript could be accepted.Text editing should be double checked (e.g.Page 1 line 61, “0.232 cm3/g” should be corrected to “0.232 cm3/g”).

Reply:

Thanks to this reviewer for accepting our work.

Also, thanks for sparing his/her valuable time reviewing this manuscript and giving fruitful suggestions.

The minor correction of cm3/g” is done.

ONCE AGAIN WE ARE HIGH THANKFUL TO THIS REVIEWER FOR SPARING HIS/HER VALUABLE TIME AND GIVING US THE FRUITFUL SUGGESTIONS.

Reviewer 2 Report

The authors didnt address all my comments.

-the removal of contaminant from water by different micro and nanostructures were reported in many papers. it suggested discussing some other structures that are used for water removal and compare with yours: https://doi.org/10.1016/j.micromeso.2021.111289, https://doi.org/10.1016/j.jhazmat.2021.128121, https://doi.org/10.1016/j.jece.2021.107091, https://doi.org/10.1016/j.cej.2021.133166

Reply:

The suggested 3 papers belong to the removal of dyes and phosphate and the 4th one is for photo-degradation. The results of these papers cannot be matched with ours as our paper deals with the removal of metal ions. However, to ACCOMMODATE TWO PAPERS we added a few lines of adsorption and materials and cite these papers.

I know that some of the references remove dye but the nanostructure platform is more important for me. I asked authors to compare their nanostructure with these refs to present insight about similar structures. but authors didnt discuss none of them.

Thus, It is suggested to properly discuss the following refs

Author Response

Pointwise replies

Manuscript ID: molecules- 1855563

Title: Polyaniline modified CNTs and graphene nanocomposite for removal of lead and zinc metal ions: Kinetics, thermodynamics and desorption studies.

First of all, I would like to thank Ms. Jacky Zhang, Section Managing Editor, to give us a chance for revising this manuscript. Besides, thanks are also the scholarly reviewer to give fruitful suggestions. Really, the incorporation of all the suggestions made this manuscript more useful and attractive to the readers. The point-wise replies to the comments of reviewers are given below.

Open Review

English language and style

( ) Extensive editing of English language and style required
( ) Moderate English changes required
( ) English language and style are fine/minor spell check required
(x) I don't feel qualified to judge about the English language and style

Yes

Can be improved

Must be improved

Not applicable

Does the introduction provide sufficient background and include all relevant references?

( )

( )

(x)

( )

Are all the cited references relevant to the research?

( )

( )

(x)

( )

Is the research design appropriate?

( )

(x)

( )

( )

Are the methods adequately described?

(x)

( )

( )

( )

Are the results clearly presented?

(x)

( )

( )

( )

Are the conclusions supported by the results?

(x)

( )

( )

( )

Comments and Suggestions for Authors

Reply:

Thanks to this reviewer for his/her appreciation of our work.

Also, thanks for sparing his/her valuable time reviewing this manuscript and giving fruitful suggestions.

The authors didnt address all my comments.

-the removal of contaminant from water by different micro and nanostructures were reported in many papers. it suggested discussing some other structures that are used for water removal and compare with yours: https://doi.org/10.1016/j.micromeso.2021.111289, https://doi.org/10.1016/j.jhazmat.2021.128121, https://doi.org/10.1016/j.jece.2021.107091, https://doi.org/10.1016/j.cej.2021.133166

 Reply:

 The suggested 3 papers belong to the removal of dyes and phosphate and the 4th one is for photo-degradation. The results of these papers cannot be matched with ours as our paper deals with the removal of metal ions. However, to ACCOMMODATE TWO PAPERS we added a few lines of adsorption and materials and cite these papers.

I know that some of the references remove dye but the nanostructure platform is more important for me. I asked authors to compare their nanostructure with these refs to present insight about similar structures. but authors didnt discuss none of them.

Thus, It is suggested to properly discuss the following refs

Second reply:

The nanostructures are hetero identities and can not be compared with one another until they have the same root. The nanostructures present in the suggested refs and the reported in this manuscript have different roots.

I have already added 2 references and even MDPI does not allow to add of more than 2 references. I know this because I am MDPI reviewer for many years.

Lastly, I am a senior Professor with a 99 h-index and 322000 citations. I  have more than 30 years' experience in WATER TREATMENT AND 20 YEARS of EXPERIENCE IN NANOTECHNOLOGY and I rarely read such comments where the reviewer is insisting unnecessarily to include references.

This is my polite observation that the suggested refs are not matching the present work and if I shall include these refs the reader will ask why I added these refs in this paper and it will reflect a bad impression on me and my authors that we don’t know how to cite refs.

This is my kind submission to the learned reviewer. 

ONCE AGAIN WE ARE HIGH THANKFUL TO THIS REVIEWER FOR SPARING HIS/HER VALUABLE TIME AND GIVING US THE FRUITFUL SUGGESTIONS.